# Surface Topography, Microbial Adhesion, and Immune Responses in Silicone Mammary Implant-Associated Capsular Fibrosis

**DOI:** 10.3390/ijms25063163

**Published:** 2024-03-09

**Authors:** Ines Schoberleitner, Leoni Baier, Michaela Lackner, Lisa-Maria Zenz, Débora C. Coraça-Huber, Wendy Ullmer, Annabelle Damerum, Klaus Faserl, Stephan Sigl, Theresia Steinkellner, Selina Winkelmann, Bettina Sarg, Daniel Egle, Christine Brunner, Dolores Wolfram

**Affiliations:** 1Department of Plastic, Reconstructive and Aesthetic Surgery, Medical University of Innsbruck, 6020 Innsbruck, Austria; 2Institute of Hygiene and Medical Microbiology, Medical University of Innsbruck, 6020 Innsbruck, Austria; 3BIOFILM Lab, Department of Orthopedics and Traumatology, Medical University of Innsbruck, 6020 Innsbruck, Austria; 4Zymo Research Corp., Irvine, CA 92614, USA; 5Pangea Laboratory, Tustin, CA 92614, USA; 6Protein Core Facility, Institute of Medical Chemistry, Biocenter, Medical University of Innsbruck, 6020 Innsbruck, Austria; 7Department of Obstetrics and Gynecology, Medical University of Innsbruck, 6020 Innsbruck, Austria

**Keywords:** implant-based breast reconstruction, SMI surface roughness, foreign body response (FBR), acute wound proteome, chronic wound proteome, antimicrobial humoral response, capsular fibrosis, surgical site microbiome, SMI-adhesive microbiome, implant-associated biofilm formation, SMI topography-specific antimicrobial response, *Staphylococcus* transmission at surgical site, immunomics

## Abstract

Breast cancer is the most common cancer in women globally, often necessitating mastectomy and subsequent breast reconstruction. Silicone mammary implants (SMIs) play a pivotal role in breast reconstruction, yet their interaction with the host immune system and microbiome remains poorly understood. This study investigates the impact of SMI surface topography on host antimicrobial responses, wound proteome dynamics, and microbial colonization. Biological samples were collected from ten human patients undergoing breast reconstruction with SMIs. Mass spectrometry profiles were analyzed for acute and chronic wound proteomes, revealing a nuanced interplay between topography and antimicrobial response proteins. *16S rRNA* sequencing assessed microbiome dynamics, unveiling topography-specific variations in microbial composition. Surface topography alterations influenced wound proteome composition. Microbiome analysis revealed heightened diversity around rougher SMIs, emphasizing topography-dependent microbial invasion. In vitro experiments confirmed staphylococcal adhesion, growth, and biofilm formation on SMI surfaces, with increased texture correlating positively with bacterial colonization. This comprehensive investigation highlights the intricate interplay between SMI topography, wound proteome dynamics, and microbial transmission. The findings contribute to understanding host–microbe interactions on SMI surfaces, essential for optimizing clinical applications and minimizing complications in breast reconstruction.

## 1. Introduction

Breast cancer stands as the most commonly diagnosed cancer in women globally, with a noteworthy part of mastectomy patients opting for breast reconstruction [1]. The use of silicone mammary implants (SMIs) in reconstruction has been prevalent since the 1960s [1,2,3]. Breast reconstruction serves multiple objectives, including reshaping the breast post-tissue loss from cancer, revising previous surgeries, and augmenting breast volume for cosmetic enhancement. The surgical implantation of biomaterial, albeit noninvasive, induces injury, triggering a fibrotic response [4,5].

Despite providing physical and psychological benefits, surgical implantation of SMIs initiates an injury-induced fibrotic response [6,7]. Silicone, a prevalent implant material, induces fibrotic responses, causing one of the most common complications of implant-based breast reconstruction or augmentation surgery, capsular contracture [8,9,10]. Remarkably, silicone continues to be the most widely used implant material in routine medical practice, despite associated side effects, including the development of capsular fibrosis, leading to pain, distinct aesthetic changes, and impaired implant function [11,12,13]. 

The encapsulation of medical prosthetic implants arises from surgery-induced injury and inflammation, activated by various factors including persistent infection, autoimmune reaction, allergic response, chemical insult, radiation, and tissue injury, which manifest through heightened extracellular matrix (ECM) synthesis [14]. Fibrosis, fundamentally a reparative mechanism, encompasses fibroblast activation and the involvement of innate immune cells [7]. Upon SMI insertion, the foreign nature of the silicone prompts an immune response characterized by inflammation and the mobilization of immune cells targeting the foreign entity. This immune cascade, orchestrated by cellular and signaling pathways like TGFβ, Smad, NF-κB, and MAPK, instigates persistent inflammatory processes that culminate in fibrotic events [6,7,15,16,17,18]. Inflammatory mediators, chiefly macrophages and neutrophils, drive profibrotic signaling pathways, fostering myofibroblast differentiation. Persistent myofibroblast activity exacerbates ECM production, culminating in the formation of collagen-I-rich fibrous matrices [6,7].

Common risk factors for capsular fibrosis include biofilm, surgical site infections, history of prior capsular contracture or fibrosis, history of radiation therapy, implant characteristics, and nonspecific protein adhesion from wound fluid and local wound tissue to implant surfaces [6]. 

Proteome studies with SMIs revealed a diverse array of proteins intricately linked to the immune response, inflammation, and wound healing [19,20,21]. By an extensive characterization of the proteomic profiles of serum, wound fluid, and SMI surfaces in patients undergoing simultaneous prophylactic nipple-sparing mastectomy with breast tissue-expander-based reconstruction, we identified a systemic burst of foreign body response (FBR) immediately after SMI implantation, marked by significant antimicrobial activity and inflammasome activation [21]. The local wound proteome, expressed in the tissue, exhibited both immediate and prolonged pro-inflammatory mediation on the SMI surface by adhesion. 

The impact of the implant’s surface texture on capsular fibrosis incidence is evident [6], and the interplay of surface chemistry and topography significantly influences protein adhesion [5,20,22,23]. SMI surfaces are categorized based on scanning electron microscopy (SEM) and surface roughness (Ra), yielding smooth (Ra < 10 µm), microtextured (10 µm ≤ Ra ≤ 50 µm), and macrotextured (Ra > 50 µm) classifications [24]. 

Initially, SMIs had a smooth surface, but due to the association with capsular contractures [6,24], macrotextured surfaces emerged. Despite reduced risks of slipping or twisting and capsular fibrosis, macrotextured surfaces are associated with breast implant-associated large cell lymphoma (BIA-ALCL) [4,25], uncommon T-cell lymphoma, and massive inflammatory reactions and chronic antigen stimulation [23,26,27,28]. Implants with low roughness, within a 2–5 µm surface roughness, resemble smooth implants but are believed to facilitate mammary fibroblast spread and reduce inflammatory responses compared to larger macrotextured surfaces in vitro and in animal models [23,29]. Patient-based investigations confirm that SMI surface roughness affects both the immediate acute and chronic early-stage fibrotic responses [30]. Reduced surface roughness, specifically to Ra 4 μm, holds promise in mitigating immune reactions, promoting healthy wound healing, and limiting excessive fibrosis, resulting in reduced capsular thickness [31]. Protein adherence to rough surfaces mediates pro-inflammatory and profibrotic processes, indicating profibrotic modulation and increased immune response [30].

Despite adherence to specific sterilization and disinfection guidelines [32,33,34], SMIs trigger an inflammatory response, recruiting immune cells to eliminate debris and potential threats. The body’s defense mechanisms against microbial invaders, particularly bacteria, play a critical role in the response to SMIs [35]. This raises the possibility of microbial colonization and biofilm formation on the implant surface, posing challenges for the immune response and antibiotic treatment [36,37,38,39,40,41]. Biofilms, protective bacterial communities encased in an extracellular matrix, can develop on the implant surface, resisting the immune response and antibiotics [38,39,41,42]. While silicone itself lacks microbial properties, the immune system releases antimicrobial peptides (AMPs) as a general defense mechanism, contributing to chronic inflammation [41,43]. Prior research indicates that breast implant bacterial contamination can manifest without clinical symptoms, leading to a chronic subclinical infection linked to capsular contracture in >10% of patients [41,44,45]. The routine use of antimicrobial pocket irrigation, implant-soaking agents, the inframammary fold incision technique, and submuscular implant placement has led to decreased rates of capsular contracture [46,47,48,49,50,51]. 

In 1981, a case report suggested a potential link between latent bacterial infection and capsular contracture [52,53,54,55]. Subsequent studies, spanning nearly two decades, further demonstrated the isolation of coagulase-negative bacteria *Staphylococcus epidermidis* from the implant or tissue in cases of capsular contracture [38,39,40,41,44,56,57,58,59]. The formation of *S. epidermidis* biofilms was investigated on differently textured silicone surfaces, revealing that rougher surfaces attract more bacteria and support thicker biofilm growth [59]. Similar findings were reported in a study with different bacterial genera (*Salmonella*, *Listeria*, and *Escherichia*), where a negative linear relationship between activation energy and surface roughness indicated easier bacterial adherence to rough surfaces [60,61,62]. By exploring sub-micrometer-sized surfaces, it was shown that surface texture, even without biocides or antibiotics, influences biofilm formation for both Gram-positive and Gram-negative bacteria [63]. Increased bacterial adherence to rough surfaces was noted, suggesting potential material-dependent variations, although the focus was on *S. epidermidis* contamination on alloys [64]. Significantly, a potential explanation for the association between textured implants and BIA-ALC suggests a higher likelihood of bacterial biofilm colonization on the surface of textured implants compared to smooth implants. This, in turn, may trigger chronic inflammation, potentially leading to tumorigenesis in susceptible individuals [65].

To date, there are no existing studies of an SMI surface-associated microbiome, or the microbiome resulting from microbial transmission and acute wound infection post-implantation to adhesion, colonization on the encapsulated SMI surface, and integration into the surrounding capsular tissue during early-stage fibrosis in real time in vivo.

Guided by the hypothesis that infection may originate from the transfer of skin microbiome during surgery, leading to biofilm formation on the implant surface which triggers chronic inflammation around the encapsulated implant, our study aims to identify, characterize, and profile microbial contamination and population profiles in the acute wound milieu and later, associated with the implant surface 6–8 months after surgery.

In pursuit of biological significance, we intraoperatively compared two silicone mammary tissue expanders (inflatable SMIs) with varying topography: (i) the conventionally used CPX^®^ 4 (average roughness radius (Ra) 60 µm; Mentor) and (ii) the novel, surface-roughness reduced device SmoothSilk^®^ (average roughness radius (Ra) 4 µm; Motiva Flora^®^). This comparison aimed to elucidate a common, as well as silicone-chemistry- and surface-roughness-exclusive, SMI-associated microbiome. Employing next-generation DNA sequencing targeting the bacterial *16S rRNA* gene, we analyzed the microbiome of the wound bed fluid (24–120 h post-op), the tissue expander that was surface-adsorbed and adhered, and the SMI-encapsulating tissue (6–8 months post-op) from breast cancer patients undergoing simultaneous prophylactic NSME and tissue-expander-based breast reconstruction. To validate the sequencing data, we performed a culture and MALDI TOF identification of skin swab samples collected from the incision site, wound bed fluid, capsular tissue, and explanted SMI biological samples obtained during implantation and removal surgery. In vitro investigations were conducted to verify our in vivo results, focusing on the colonization, growth, and biofilm formation of *S. epidermidis* and *S. aureus* on sterile smooth, low-textured, and rough SMI surfaces. The latter corresponded to the devices used in previous patient studies. Finally, the immunoreactivity of in vitro inoculated silicone patches was determined by analyzing the gene expression of various inflammatory markers on biofilm-covered SMI patches cocultured with peripheral blood mononuclear cells (PBMCs).

Our findings offer unprecedented insights into intrasurgical wound infection, microbiome transmission and adhesion onto SMI surfaces, and integration into the capsule, in real time both in vivo and in vitro. Additionally, we identified potential antimicrobial biomarkers in SMI-associated capsular fibrosis, offering insights into novel therapeutic targets and enhancing our understanding of the causal relationship between SMIs, biofilm formation on medical prosthetics, and the autoimmune response of the immune system.

## 2. Results

### 2.1. Impact of Surface Roughness on Antimicrobial Response and Wound Proteome Adhesion on SMI Surface

Our previous work elucidated the three-dimensional composition of the surface-associated proteome of SMIs, encompassing adhered plasma, local tissue-derived proteins in the acute wound, and those expressed in early fibrosis stages [21]. 

To explore the effects of reducing implant surface roughness from Ra 60 µm (SMI 60 µm; CPX^®^4, Mentor LLC, Johnson & Johnson Medical GmbH, Germany) to Ra 4 µm (SMI 4 µm; SmoothSilk^®^, Motiva Flora^®^, Establishment Labs, Costa Rica) on antimicrobial inflammatory proteome response in the acute wound and long-term adhered to the expander surface, we turned to our previously generated mass spectrometry profiles of the collected wound proteome data [21]. This dataset covered plasma and acute wound proteome profiles (24–120 h post-op) and the surface-associated proteome (6 to 8 months post-op) to both SMI types, allowing us to compare protein distributions. During the first five days post-SMI-implantation, the acute wound proteome resembled an inflammatory storm with significant involvement of antimicrobial agents (AMPs) that partially, in the course of 6 months, adhere to the SMI surface [21]. Sixty-five of the inflammatory molecules identified in the acute wound were involved in the antimicrobial humoral response (Appendix A). Of note, we found no topography-specific antimicrobial response in the acute wound in the first five days post-implantation.

For deeper insights into the functions of the 65 identified AMPs in WBF, we conducted gene ontology (GO) molecular function enrichment analysis (Figure 1a) that indicated associations of 12 proteins with the role of a cell wall digestive autolysin or inhibitors of peptidoglycan synthesis. 

Subsequently performed STRING database [66] analysis of known and predicted physical and functional protein–protein interactions (Reference NCBI taxonomy Id: 9606) generated a protein–protein interaction (PPI) regulatory network of the 12 candidate proteins (cell wall autolysis, inhibition of peptidoglycan synthesis) with 12 nodes and 17 edges, average node degree 2.83, average clustering coefficient 0.537, expected edge number 1, and PPI enrichment *p*-value of 3.24 × 10^−14^ (Figure 1b, left). Strikingly, the edges represent only one predicted functional association of the 12 nodes (12 candidate proteins) with the antimicrobial activity (Figure 1b, right: reactome pathways). Specifically, the heatmap visualization (Figure 1c) with applied Manhattan clustering of wound proteome differentiation over the first five days after expander implantation revealed a chronological topography-specific reduction in wound proteome composition and abundance on day 5, reflecting a closer relation to presurgery plasma proteome. Interestingly, previously, we identified two clusters: an immediate inflammatory storm with wound bed fluid (WBF) proteomes formed around SMI 4 µm at 24 h and 48 h post-operation and around SMI 60 µm at 24 h post-operation [21], and the second cluster encompassing later stages with chronologically reduced abundances of the 12 AMPs. We conclude that the potential for activation and secretion of these 12 AMPs persists for a more extended period around the smoother device. 

Thus, our comprehensive investigation into the three-dimensional composition of the surface-associated proteome of the SMI 24–120 h post-op and the subsequent exploration of the impact of surface roughness reduction on antimicrobial inflammatory proteome response unveiled a nuanced interplay between topography, proteome dynamics, and the prolonged activation of antimicrobial responder proteins.

Five of sixty-five AMPs found in wound bed fluid adhered to both SMI surfaces six months post-operation (Figure 2a and Appendix A), with plasma serving as the primary source for antimicrobial response agents [21]. Notably, FLG2 [67,68,69,70], part of the antimicrobial inflammatory response and profibrotic driver from the S100A family, was present in WBF formed around both devices, but it adhered exclusively to the rougher surface (Figure 2b and Appendix A), confirming a topography-exclusive chronic antimicrobial inflammatory response that additionally drives fibrosis and implant encapsulation around rougher implants 6–8 months post-implantation.

We postulate that during surgery, an infection may occur in the patient, facilitated by the transfer of skin microbiome to lower tissue layers and onto the implant surface through the operative incision and implant placement. This process leads to the formation of biofilm on the implant surface, initiating chronic inflammation around the encapsulated implant. This, in turn, propels the inflammatory response, consequently contributing to fibrosis and thickening of the capsule.

### 2.2. Quantification and Data Integrity of Intraindividual Comparative Microbiome Profiling in Wound Bed Fluid, Capsular Tissue, and SMI-Adhesive Microbiome

This clinical investigation addressed a crucial knowledge gap in understanding the impact of SMI surface texture on microbial surface adhesion and biofilm formation and acute and chronic immune responses in patients undergoing prophylactic nipple-sparing mastectomy (NSME) and SMI-based breast reconstruction. During surgery, we intraoperatively compared two types of tissue expanders, the CPX^®^ 4 (termed from here on SMI 60 µm, roughness radius: 60 µm Ra; Mentor) and SmoothSilk^®^ (termed from here on SMI 4 µm, roughness: 4 µm Ra; Motiva), which differ in surface topography. Blood and wound bed fluid samples were collected at 24 h, 48 h, 72 h, 96 h, and 120 h post-tissue-expander implantation. Additional blood and capsular tissue samples were collected during reoperation upon expander removal and exchange with a definitive implant. Immediately following tissue-expander removal, the expander surface was swabbed and stripped to collect additional samples for microbiome testing.

In collaboration with Pangea Laboratory (Tustin, CA, USA) and Zymo Research Corp. (Irvine, CA, USA), next-generation DNA sequencing (NGS) was performed to characterize the microbiome in acute wound bed fluid (24–120 h post-op), SMI surfaces, and capsular tissue (6–8 months post-op) and to identify bacteria that may trigger the antimicrobial response postsurgery [21]. The samples of wound bed fluid formed around the SMI, SMI surface swabs, and capsular tissue specimens were comparatively analyzed to understand the antimicrobial contamination of the acute wound and SMI during surgery and the first five days post-implantation; the microbial surface adhesion; and the biofilm formation on the SMI as well, as its incorporation into the encapsulating fibrotic tissue 8 months post-op. WBF, SMI surface, and capsular tissue sample data were normalized to negative controls (a) not associated with implants (surgery room atmosphere, DNA/RNA Shield™ reagent used for sample stabilization), (b) negative controls for SMI 60 µm and SMI 4 µm (WBF drainage container valve, “sterile” implant packing, tissue prep area under cell culture flow, scalpel for tissue prep), and negative controls (Appendix A) for the capsular tissue processing procedure (Host Zero).

Following normalization to negative controls, 501 total species representing 414 genera from 140 families were identified from the three specimen types (WBF, SMI surfaces, and capsular tissue) by NGS. Similar counts of bacterial families were identified in samples from the different expander types (SMI 4 µm and SMI 60 µm) for each specimen type (Figure 3a and (Appendix A)). The most commonly observed taxa in the capsular tissue specimens for both the SMI 4 µm (*n* = 5) and SMI 60 µm (*n* = 4) expander surfaces was *Pelomonas saccharophila*, in the Comamonadaceae family, previously identified as a commensal of the dermis [71]. For the wound bed drainage, the most commonly observed families identified in specimens from samples collected from both expander surface types included the Comamonadaceae family (*n* = 26 for SMI 4 µm and SMI 60 µm), owing largely to the presence of *Ralstonia* sp., which has been isolated from numerous water sources [72]. Staphylococcaceae were the next most abundant family, identified from 23 and 22 WBF specimens for the SMI 4 µm and SMI 60 µm expander types, respectively. *Staphylococcus hominis* was identified in 17 WBF specimens from the SMI 4 µm and 11 from the SMI 60 µm expander surface, while *S. epidermidis* was detected in 13 WBF specimens collected from each of the expander types. 

Bacterial communities were further explored by comparing community structure across the three different specimen types originating from the two expander types. Local (alpha) diversity analysis, a measure of taxa richness, was observed to be consistent in the 5 days post-tissue-expander implantation and in the SMI surface swabs and was lowest in the capsular tissue specimens (Figure 3b). No significant differences in taxa richness were detected between the two expander types. The diversity between the communities (beta diversity) originating from the two expander types was explored by calculating Bray–Curtis dissimilarity and visualized by a principal component plot (Figure 3c). Points, each representing a specimen, appeared to cluster according to specimen type, though no clear separation was observed between the SMI 4 µm and SMI 60 µm expander types and no significant differences were observed by permutational analysis of variance.

### 2.3. Microbial Dynamics in the Peri-Implant Environment: A Comprehensive Analysis of Skin Microbiome Transfer and Biofilm Formation during Implant-Based Breast Reconstruction

To assess our hypothesis that transfer of the skin microbiome from the incision site to the implant surface leads to microbial surface adhesion and biofilm formation in patients undergoing breast reconstruction (Figure 4a), we conducted a comparative analysis of microbes found in wound bed fluid, the SMI surface, and the intracapsular region within the same individuals, with specific reference to the SMI topography. The feasibility of interindividual comparison was precluded by the inherent variability in microbiome composition observed between patients.

In the acute wound 24–120 h post-implantation, principal component analysis revealed significant topography-specific sample variation, with principal component 1 and principal component 2 explaining 15.6% and 17.4% of the total variance and revealed segregation between the two differential wound bed fluids formed around SMI 4 µm and SMI 60 µm at every sampling time point, respectively (Figure 4a). The microbiome of the acute wound around SMI 4 µm exhibited a significant difference compared to SMI 60 µm at all sampled time points 24–120 h post-implantation. In WBF, a collective count of 33 bacterial families was observed around SMI 4 µm, while 111 were identified around SMI 60 µm. A total of 30 bacterial families were common to both SMIs, 3 exclusive to the SMI 4 µm surface, and 81 exclusive to SMI 60 µm (Figure 4b), all corroborated in Appendix A. Specifically, the temporal visualization of acute wound microbiome differentiation over the first five days after SMI implantation revealed a periodical time-dependent compositional shift, as indicated by an immediate increase in Actinobacteria abundance on day 1, followed by a gradual regressive correlation with the time point post-op until day 3. On day 4, an enrichment of Firmicutes was observed in the WBF formed around SMI 4 µm, whereas it was observed around the rougher surface on day 5, indicating a time-point-dependent alteration in the microbiome composition dynamics due to topography. Gemmatimonadetes and Fusobacteria were detected only around SMI 4 µm, contributing to a differential composition profile on all five days. The acute wound microbiome composition is topography-dependent.

PCA analysis of the SMI surface microbiome 6 to 8 months post-implantation (Figure 4c) highlighted the spatial distribution of the 14 samples (7 swabs each of the 4 µm and the 60 µm textured SMI surface 6–8 months post-op) and revealed segregation between the two differential SMI surfaces intra- and interindividually in all patients. A total of 33 bacterial species were observed adhered to SMI 4 µm, while 38 were identified adhered to SMI 60 µm (Appendix A). Seventeen bacterial families were commonly adhered to both SMIs (Figure 4d). 

During reoperation, capsular tissue (approx. 3 × 3 cm) was harvested from both implants at two positions, the anterior contact zone with TiLOOP^®^ and the posterior (TiLOOP^®^-free) contact zone with *M. pectoralis*. PCA analysis revealed a significantly different microbiome composition incorporated into the capsular tissue (6–8 months post-op) around SMI 4 μm compared to the rougher device. The intracapsular microbiome composition was not affected significantly by the proximity of a titanium-coated polypropylene mesh (Figure 4e). A total of 5 intracapsular bacterial families were observed in the capsule around SMI 4 µm, while 13 were identified in the capsule around SMI 60 µm (Appendix A). Four bacterial families commonly invaded the capsule around SMIs (Figure 4f). While Corneybacteriaceae was the only family exclusively detected in the capsule around SMI 4 µm, we found nine bacterial families that infiltrated the intracapsular region around SMI 60 µm. Staphylococcaceae were identified around both devices in the wound fluid (Figure 4b) 24–120 h post-op, attached to the surface (Figure 4d), and incorporated into the capsule (Figure 4f) 6–8 months post-op.

Collectively, our data highlight microbial invasion of the acute wound, transfer and adhesion to the SMI surface, and intracapsular invasion in the chronic wound 6–8 months post-op in breast implant-based breast reconstruction. Moreover, we demonstrate the impact of silicone implant surface topography alterations on wound microbiome composition. We detected higher microbial diversity and quantity around the rougher device that could play a role in the increased inflammatory and profibrotic response compared to the Ra 4 µm surface roughness, demonstrated recently [21,30,31].

NGS analysis of the SMI-associated microbiome does not differentiate between DNA from live or dead cells. To determine whether live bacteria were present, wound bed fluid was culture-tested, but the culture was sterile. We were unable to determine whether microbial DNA in the sample was from unculturable live or dead cells.

### 2.4. Staphylococcal Transmission in Surgery: Skin Microbiota Transfer, Biofilm Formation, and Chronic Inflammation Leading to Implant Encapsulation

To further substantiate our hypothesis regarding patient infection during surgery through transfer of skin microbiota, we assessed the skin microbiome at the bilateral incision/implantation sites. Skin swab samples from the submammary fold and armpit, along with the back of the neck as a negative control, were cultured and analyzed using the MALDI-TOF method.

The microbiota detected in swab samples were low diversity. In total, eight species were identified (Figure 5). A total of 50% of the patient samples included Staphylococcus hominis and 25% included Staphylococcus epidermidis as part of their underbust, armpit, and neck area skin microbiome. 

The limited microbial diversity in skin swab samples suggests a potential association between specific microbial species and the surgical site, emphasizing the importance of understanding individual skin microbiomes in the context of surgical procedures. For example, *Staphylococcae* were identified in acute and chronic wounds and were associated with the SMI surface. Further research and larger sample sizes are warranted to establish a comprehensive understanding of the influence of microbial dynamics and implant-surface-associated biofilms on foreign body response, surgical outcomes, and complications in implant-based breast reconstruction and will provide valuable insights for infection prevention strategies in clinical settings.

### 2.5. In Vitro Evaluation of Silicone SMI Surface Topography Impact on Staphylococci Adhesion, Growth, Colonization, and Biofilm Formation

To evaluate SMI surface impact on microbial surface adhesion and colonization, patches with diverse topographies (MENTOR^®^ smooth, CPX^®^4 textured, SmoothSilk^®^ low-textured) and roughness (Ra 0 µM, Ra 4 µm, Ra 60 µm) were inoculated in vitro with Staphylococcus epidermidis (Figure 6) and Staphylococcus aureus (Appendix A), part of the normal human microbiota, typically the skin microbiota, and less commonly the mucosal microbiota [57,73]. 

Both S. epidermidis (Figure 6a) and S. aureus (Appendix A) exhibited adherence and colonization on all silicone surfaces tested. However, the silicone patch markedly impaired the growth and colonization of both staphylococci compared to control without a silicone patch, as indicated by Figure 6b and Appendix A (**** *p* < 0.0001).

Notably, the growth and colonization of both *S. epidermidis* and *S. aureus* are significantly affected by the topography of the silicone patch (Figure 6b; * *p* = 0.0327 and Appendix A; * *p* = 0.0292). An evident difference emerges between the smooth surface and 60 μm, showcasing increased growth of *S. epidermidis* with enhanced texture (Figure 6b, * *p* = 0.0330; and Appendix A, * *p* = 0.0306). Although there is minimal contrast in *S. epidermidis* growth between smooth and 4 μM (*p* = 0.5697), a positive trend is apparent when comparing 4 μm to 60 μm (^ns^
*p* = 0.0646). This trend is similarly confirmed for *S. aureus* (^ns^
*p* = 0.07). Finally, Pearson analysis revealed a positive correlation between surface topography and Staphylococci colonization (Figure 6c and Appendix A)—the more textured, the better the growth of the bacteria.

Biofilm formation of both species serves as the key virulence factor linked to disease, as evidenced by animal models of infections related to biomaterials [57,73]. To investigate the impact of the SMI surface on bacterial biofilm development, silicone patches were examined using electron microscopy, both untreated (Figure 7a) and after inoculation with *S. epidermidis* (Figure 7b) or *S. aureus* (Appendix A) overnight. The untreated control SMI patches remained sterile (Figure 7a). We compared the external surface of the implant shell with varied topographies (smooth, 4 µm, and 60 µm) and used the internal surface of the smooth implant as a control (Figure 7b and Appendix A). Biofilm formation was confirmed on both textured surfaces, but it was notably more complex on Ra 60 µm. Singular cells and colonies were observed on the outer surface of the smooth implant shell, while a robust biofilm was evident on the inner surface of the smooth implant shell, indicating a potential risk for biofilm formation at implant rupture.

In essence, these findings confirm adhesion, growth, colonization, and biofilm formation of Staphylococci on SMIs, and these are significantly influenced by the texture of the silicone patch in vitro.

### 2.6. Immunoreactivity of Staphylococci Biofilms on Silicone Implant Surfaces In Vitro

To investigate and confirm the proteomic data obtained in vivo, whether biofilm-associated SMI patches elicit an inflammatory immune response and to assess the impact of reducing implant surface roughness from Ra 60 µm to Ra 4 µm to smooth on the inflammatory response, we incubated both staphylococci, S. epidermidis and S. aureus, biofilm-covered patches with freshly isolated peripheral blood mononuclear cells (PBMCs).

Multiple bacterial genes orchestrate biofilm formation. Bacterial stress activates sigma B (sigB), a gene regulator, which, in turn, controls staphylococcal accessory regulator A (sarA). SarA regulates polysaccharide intercellular adhesin (PIA) formation icaA [74], serving as a biomarker for biofilm [75]. The icaADBC genes govern PIA production [76], with sarA directly activating the intracellular adhesion ADB and C (icaADBC) locus and modulating superoxide dismutase expression [77]. In *S. aureus*, sodA regulates superoxide dismutase, crucial for detoxifying reactive oxygen species and minimizing cellular stress [77].

We compared the bacterial gene expression of biofilm markers icaADBC, SarA, SigB, and SodA to identify the triggering factors (Figure 8a,b) and examined human PBMC gene expression of pro-inflammatory and profibrotic markers interferon δ (IFNδ), tumor necrosis factor α (TNFα), transforming growth factor β (TGFβ), interleukin 1 β (IL1b), and interleukin 17 (IL17) (Figure 8c,d) to understand the corresponding antimicrobial response [78,79,80,81]. 

Strikingly, SEM analysis of biofilm formation on silicone patches was confirmed by analysis of real-time expression of biofilm markers associated with marker genes IcaA-D, SarA, SIgB, and SodA [82]; moreover, heightened SMI surface texture correlated with increased bacterial gene expression (Figure 8a,b). Additionally, the results unveiled a positive correlation between the immunoreactivity of the SMI surface and its topography in vitro, wherein heightened texture corresponded to an elevated antimicrobial pro-inflammatory response (Figure 8c,d).

Thus, our data show that SMI surfaces are suitable for bacterial colonization and biofilm formation in vitro and in vivo. We confirm increased colonization, growth, and biofilm formation on more textured SMI surfaces, leading to higher antimicrobial immunoreactivity in acute and chronic wounds. 

## 3. Discussion

Antimicrobial immune responses refer to the body’s defense mechanisms against microbial invaders, such as bacteria. In the context of silicone breast implants, the immune response is typically triggered by the presence of foreign material (silicone) in the body that involves the activation of immune cells and processes aimed at removing or isolating the foreign material [83,84]. The initial response to SMIs includes an inflammatory reaction [7]. Inflammation is a part of the immune response and involves the recruitment of immune cells to the implant site to clear debris and potentially harmful substances [6]. Although implanted under specific sterilization and disinfection guidelines [33,34,85], microbial colonization and biofilm formation on the implant surface can occur. Biofilms are communities of bacteria encased in a protective matrix, making them resistant to the immune response and antibiotics [38]. The antimicrobial immune response may be activated as a part of the overall immune reaction [6]. While silicone itself is not a microbial agent, the immune system may respond to the implant by releasing antimicrobial substances as a general defense mechanism. This can contribute to chronic inflammation. 

From our previously published proteome study, we discovered a diverse array of proteins intricately linked to the immune response, inflammatory processes, and facilitation of wound healing within the proximal vicinity of the SMI [21]. We extensively analyzed the surface-associated proteome of the SMI, investigating proteins from plasma, local tissue, and early fibrosis stages [1]. To assess the impact of reducing implant surface roughness from average roughness (Ra) 60 µm to Ra 4 µm on the antimicrobial inflammatory proteome response, here, we revisited mass spectrometry profiles covering 24–120 h post-op and 6–8 months post-op for both SMI types (SMI 60 µm and SMI 4 µm implanted intraindividually). The acute wound proteome exhibited an inflammatory storm within the first five days post-implantation, featuring antimicrobial agents that adhered to the SMI surface over the next 6 to 8 months [21,30]. 

Notably, 65 plasma-derived components associated with the acute wound inflammatory response played a role in the antimicrobial humoral response (Figure 1), with no topography-specific antimicrobial response initially. Deeper insights into the 65 identified AMPs in WBF revealed associations with cell wall digestive autolysin or peptidoglycan synthesis inhibitors and highlighted a regulatory network with antimicrobial activity as the predicted functional association. The investigation uncovered a nuanced interplay between topography, proteome dynamics, and the prolonged activation of antimicrobial response proteins. 

Five of the sixty-five AMPs were present in WBF around both SMI surfaces six months post-operation (Figure 2). Notably, FLG2, a participant in the antimicrobial inflammatory response and a profibrotic driver [67,68,69,70,86,87], is associated exclusively with the rougher SMI surface, confirming a topography-exclusive chronic antimicrobial inflammatory response driving fibrosis and implant encapsulation around rougher implants 6–8 months post-implantation. This substantiates our previous findings of the significant impact of surface roughness on acute inflammatory responses, fibrinogen accumulation, and the subsequent fibrotic cascade and capsular composition [30,31]. The precise technique employed for sample collection and the chosen analytical methods constitute a crucial step in establishing biological significance during the diagnostic research of capsular fibrosis etiology.

To identify the triggers of the antimicrobial immune response, we further focused on comparing the composition of the wound-associated microbiome in WBF (24–120 h post-op), adhered onto SMI surface, or integrated into the fibrous capsule (6–8 months post-op). NGS analysis was performed to characterize the microbiome and identify bacterial antigens targeted by the antimicrobial immune reaction. The acute wound phase, spanning 24–120 h post-implantation, unveiled a significant topography-specific variation in microbial composition. This topography-dependent microbial diversity was particularly pronounced in wound bed fluid (WBF), with SMI 60 µm exhibiting a notably higher count of bacterial families compared to SMI 4 µm. The temporal dynamics revealed a distinct compositional shift in the acute wound microbiome, emphasizing the role of topography in shaping microbial adhesion.

Extending our analysis to the chronic phase (6–8 months post-op), the spatial distribution of SMI surface microbiome highlighted segregation between SMI 4 µm and SMI 60 µm. Intriguingly, the capsular tissue microbiome composition reflected a significant divergence based on SMI topography. While both surfaces attracted common bacterial families, the rougher SMI 60 µm demonstrated higher microbial diversity and quantity. This finding links higher microbial diversity to increased inflammatory and profibrotic responses, aligning with previous observations [21,30]. 

The unique identification of bacterial families in the intracapsular region further accentuates the microbial invasion in the chronic wound. Staphylococcae, identified in acute and chronic wounds, surface adhesion, and capsular tissue, emphasize their persistent presence and potential role in the implant encapsulation process. Notably, differential family compositions around SMI 4 µm and SMI 60 µm (Figure 4) suggest nuanced interactions with the implant’s topography, impacting the microbial landscape within the capsule.

Here, we confirm that the reduction in SMI surface roughness to an average of 4 μm emerges as a promising approach for mitigating detrimental immune reactions, promoting healthy wound healing, and curbing excessive fibrosis [21,30,31]. 

Our results highlight microbial invasion of the acute wound, transfer to the SMI surface, and intracapsular invasion in the chronic wound [38,41,88,89], demonstrating not only the impact of silicone implant surface topography alterations on wound microbiome composition, but also emphasizing the role of microbial dynamics in implant-based breast reconstruction. 

The limitations of NextGen sequencing for detecting living cells were acknowledged, prompting the need for additional analyses such as Staphylococcae culture to confirm microbial presence. 

To substantiate the hypothesis of skin microbiota transfer during surgery, we assessed skin microbiomes at incision/implantation sites. Only eight species were identified, with *Staphylococcus hominis* and *Staphylococcus epidermidis* being prevalent in 50% and 25% of patients, respectively (Figure 5). The limited microbial diversity suggests a potential association between specific microbial species and the surgical site. Notably, Staphylococcae, identified in acute and chronic wounds, were also associated with the SMI surface (Figure 4). In aesthetic breast surgery, generally categorized as clean surgery, studies of postoperative surgical site infection (SSI) rate increase, commonly identified bacteria in these infections include *Staphylococcus*, *Escherichia*, *Pseudomonas*, *Propionibacterium*, and *Corynebacterium* [41,90,91,92,93]. Of note, *staphylococci* are the most common axillary flora, and antibiotics targeting them do not significantly impact SSIs [36,94]. Our findings underscore the significance of understanding individual skin microbiomes in the context of surgical procedures. Limited diversity in skin swab samples underscores the need for larger-scale investigations to establish comprehensive associations between specific microbial species and surgical outcomes. The absence of live bacteria in culture tests poses intriguing questions about the origin and nature of microbial DNA in the samples. This knowledge is crucial for comprehending the dynamics of microbial interactions during implant-based breast reconstruction surgeries.

While this study provides valuable insights, we highlight the need for further research with larger sample sizes to establish a comprehensive understanding of microbial dynamics. This is particularly relevant for assessing the impact of microbial interactions on foreign body response, surgical outcomes, and complications in implant-based breast reconstruction and could contribute to more effective preventive measures.

Moving to an in vitro setting, the evaluation of SMI surfaces patches with diverse topographies revealed significant effects on microbial adhesion, growth, and colonization by *S. epidermidis* and *S. aureus* (Figure 6 and Appendix A). Both bacterial species exhibited adhesion and colonization on all surfaces tested. Despite adherence and colonization occurring on all silicone surfaces, the presence of silicone markedly inhibited the growth and colonization of both *Staphylococci* species compared to controls without silicone patches. Importantly, the topography of the silicone patch significantly influenced the growth of bacteria, with increased texture correlating positively with enhanced bacterial colonization. 

Biofilm formation (Figure 7 and Appendix A) is a recognized virulence factor linked to diseases, especially in the context of biomaterial-related infections. The findings presented in this study, as evidenced by electron microscopy imaging, shed light on the intricate relationship between SMI surfaces and bacterial behavior. The electron microscopy analysis of untreated SMI patches (Figure 7a) revealed that the control surfaces remained sterile, indicating that biofilm formation did not occur spontaneously on the silicone material. However, after inoculation with *S. epidermidis* (Figure 7b) or *S. aureus* (Appendix A) overnight, biofilm formation was evident on both textured surfaces. Biofilm complexity was more pronounced on surfaces with a roughness of Ra 60 µm. Notably, the topography of the silicone patch significantly influenced bacterial growth and colonization. Enhanced texture correlated with increased bacterial growth, emphasizing the importance of surface topography in modulating microbial behavior on implants.

The comparison between the external and internal surfaces of the smooth implant shell provided additional insights. Singular cells and colonies were observed on the outer surface, while a robust biofilm was present on the inner surface. This observation raises concerns about the potential risk for biofilm formation at the site of implant rupture, emphasizing the importance of understanding how surface characteristics influence bacterial behavior.

These findings underscore the intricate interplay between SMI surface characteristics and microbial behavior, emphasizing the importance of surface texture in influencing bacterial adhesion and biofilm formation [38,39,42,43]. Further insights from this study contribute to our understanding of microbial interactions with SMI surfaces and have implications for the development of implants with optimized properties to mitigate microbial-related complications.

Crucially, in vitro analysis of *S. epidermidis* and *S. aureus* biofilm-covered patches incubated with peripheral blood mononuclear cells (PBMCs) revealed a heightened expression of biofilm marker *IcaA-D*, *SarA*, *SIgB*, and *SodA* [82] genes, confirming biofilm formation on SMI surfaces (Figure 6). Strikingly, the increased surface texture of SMIs correlated positively with elevated bacterial gene expression, highlighting the significance of topography in influencing microbial responses. The positive correlation between SMI surface immunoreactivity and topography not only confirms the role of surface texture in triggering an enhanced antimicrobial pro-inflammatory response in vitro but substantiates the integrity of the presented proteomics and microbiomics analysis in vivo in human patients. 

Our findings not only affirm the suitability of SMI surfaces for bacterial colonization and biofilm formation in vitro and in vivo but also underscore the heightened colonization, growth, and biofilm formation, coupled with increased antimicrobial immunoreactivity, particularly on more textured SMI surfaces. 

This comprehensive investigation into the intricate interplay between antimicrobial immune responses and silicone breast implant (SMI) surface characteristics highlights the necessity of contemplating potential avenues for future research to optimize clinical outcomes in breast reconstruction. An intriguing trajectory involves the examination of the impact of antibiotic prophylaxis, surgical techniques, or implant coatings on microbial colonization and host responses. Future investigations should scrutinize the effectiveness of diverse prophylactic measures in mitigating infection and fibrotic responses, providing a foundation for enhanced strategies in minimizing complications during breast reconstruction surgeries. A nuanced analysis of how silicone implant surface characteristics interact with various preventive interventions could unravel novel insights into modulating immune reactions and promoting healthier wound healing.

To propel our comprehension forward, future studies must encompass larger and more diverse cohorts, ensuring a thorough exploration of these factors’ influence on microbial dynamics and clinical outcomes over extended follow-up periods. This multifaceted approach is indispensable for unraveling the complexities of breast implant-related complications and developing targeted interventions to augment the safety and efficacy of silicone breast implants in clinical applications.

Furthermore, our study unfolds avenues for further investigation into the impact of antibiotic prophylaxis, surgical techniques, and implant coatings on microbial colonization and host responses. Examining the effectiveness of different prophylactic measures in mitigating microbiome-associated complications, optimizing surgical techniques to minimize microbial transmission, and developing advanced implant coatings with antibiofilm properties could markedly enhance clinical outcomes in breast reconstruction.

Moreover, understanding the role of specific antimicrobial biomarkers identified in SMI-associated capsular fibrosis could unveil novel therapeutic targets, advancing our comprehension of the causal relationship between SMIs, biofilm formation, and immune responses. These promising directions bear the potential not only to improve patient outcomes but also to reduce complications in breast implant-based reconstruction procedures, marking a significant stride in refining and advancing breast reconstruction practices.

Our study on intrasurgical wound infection, microbiome transmission, and adhesion onto silicone breast implant (SMI) surfaces in breast implant-based reconstruction provides valuable insights but acknowledges several important limitations. The variation in sample sizes is acknowledged as a notable limitation, introducing the potential for bias and impacting the generalizability of the findings. Patient-specific factors influencing microbial composition pose inherent challenges, emphasizing the need for a standardized baseline for comparison. The dynamic nature of microbial populations and the complexity of host–microbiome interactions underscore the importance of conducting longitudinal studies to validate the observations over an extended follow-up period.

Additionally, this study’s focus on specific types of silicone mammary tissue expanders with varying topography may limit generalizability to other implant types or materials. The exclusive examination of surface-adsorbed and adhered microbiomes may not fully capture the complete spectrum of microbial dynamics, and the inability to differentiate between live and dead cells in NGS analysis raises questions about the viability of the detected bacteria.

Despite efforts to validate sequencing data through culture and MALDI TOF identification, in vitro investigations might not fully replicate the complexity of the in vivo environment. The intriguing correlation between silicone implant topography and the inflammatory response, while promising, requires further validation in larger patient cohorts and diverse clinical settings.

In conclusion, while shedding light on the intricate relationship between SMI surface characteristics, microbiome dynamics, and the host’s immune response, this study highlights the need for careful consideration of its limitations. Future research should address these constraints by employing larger sample sizes, diverse implant types, and extended follow-up periods to enhance the robustness and applicability of the findings in the clinical context. Providing valuable insights into host–microbe interactions on implant surfaces, these findings play a crucial role in guiding strategies to minimize complications and optimize the design of SMIs in clinical applications.

## 4. Materials and Methods

### 4.1. Study Population

As briefly described in Schoberleitner et al. [21,31], this study involved the enrollment of ten female patients undergoing bilateral prophylactic mastectomy and simultaneous tissue-expander-based breast reconstruction due to high-risk hereditary predisposition and/or confirmed *breast cancer gene 1* and/or *breast cancer gene 2* (*Brca1+/Brca2+*) mutation. Informed consent for photo documentation, the operation, sample collection, and anonymized evaluation and publication of data was obtained in written form from all patients after confirmation of all inclusion and exclusion criteria (Table 1). 

The Expander-Immunology Trial, registered under ClinicalTrials.gov ID NCT05648929, implemented the implantation of two distinct tissue expanders, CPX^®^4 (MENTOR LLC, Johnson & Johnson Medical GmbH, Germany: from here referred to as SMI 60 µM) and SmoothSilk^®^ (Motiva Flora^®^, Establishment Labs, Costa Rica: from here referred to as SMI 4 µm), characterized by varying surface topographies. Notably, the selection of tissue expanders for each patient’s left or right breast was randomized.

This study encountered withdrawals and exclusions, with one patient withdrawing due to a histological breast cancer diagnosis in one mastectomy sample and two patients being excluded due to postoperative complications. Consequently, seven patients were included in the evaluation. Informed written consent was obtained from all participants, covering photo documentation, surgical procedures, sample collection, and the anonymized evaluation and publication of data.

All donor biological samples (blood, wound bed fluid, capsular tissue, and removed tissue expander) and associated information were acquired in strict adherence to the regulations of the Institutional Ethical Committee of the Medical University Innsbruck, Austria, the Declaration of Helsinki, and The European Union Medical Device Regulation (§40 Section 3 Medical Devices Act). The analysis encompassed 14 peripheral blood draws, 70 wound bed fluid samples, 28 capsular tissue specimens, and 14 tissue-expander surface strips and swabs, providing comprehensive insights into the immunological aspects of the breast reconstruction process. Further details regarding the study population can be referenced in Schoberleitner et al. [21,30,31]. 

### 4.2. Study Design

This monocentric, randomized, double-blind controlled clinical study was approved by the Institutional Ethical Committee of the Medical University Innsbruck, Austria (protocol code 1325/2019, 23 January 2020) and the Austrian Federal Office for Safety in Health Care (approval number; 13340962). To prevent the detection of device-specific immune reactions or microbiome compositions and to analyze an SMI topography-dependent host response and microbial transfer, we opted to implant two tissue expanders, both comprising a poly(dimethyl siloxane) (“PDMS”) elastomer shell, with diverse surface topographies. We evaluated a total of 7 patients, who received either the routinely used expander Mentor CPX™4 (termed SMI 60 µm) or the novel Motiva SmoothSilk^®^ with reduced surface topography roughness (termed SMI 4 µm). The allocation to the left or right breast after prophylactic bilateral NSME was randomized, and both patients and laboratory experts were double-blinded. Matching was conducted intraindividually and interindividually based on the implanted tissue expander. The inflatable tissue expanders were subsequently replaced with definitive implants in a second surgery occurring 6 to 8 months post-implantation. For an extensive study design description, refer to Schoberleitner et al. [21,30,31].

### 4.3. Biological Sample Collection

The blood draws were conducted concurrently with anesthesia before both the initial tissue-expander implantation and the subsequent tissue-expander removal/exchange with a definitive implant. Biological samples of wound bed fluid (referred to as WBF) were collected daily 24, 48, 72, 96, and 120 h following expander implantation. Wound drains, integral to the surgical procedure for patients undergoing expander-based reconstruction, were retained postoperatively. WBF was collected under sterile conditions in sterile containers at room temperature. For the initial 24 h, no vacuum was applied to the drains. However, after this period, the drains were maintained with a vacuum until removal. Flasks containing WBF were removed every 24 h within the timeframe of 24 to 120 h postoperatively, representing the total collection time of 120 h. For microbiome profiling by NGS, portions of the samples were stabilized with DNA/RNA Shield reagent stabilization solution (Zymo Research Corp., Irvine, CA, 92614, USA) and aseptically frozen at −80 °C. As a control for sample acquisition from the first surgery (SMI implantation), we stored swabs of the surgery room airspace, WBF container tubes, and a specimen of the DNA/RNA stabilization reagent. For primary cultures attempts, aliquots were stored in RPMI media (Gibco). To isolate proteinaceous fractions from peripheral blood and WBF, we employed gradient separation of drain fluid using Ficoll-Paque^®^ (Cytivia) to eliminate the cellular component. The proteinaceous WBF and plasma were subsequently sterilized by passing it through a 0.1 µm and then a 0.07 µm syringe filter to remove all cells, both human and microbial. The resulting proteinaceous fraction was frozen at −80 °C for subsequent processing using a TMT-based quantitative proteomic approach and immunoassays.

During reoperation, capsular tissue (approx. 3 × 3 cm) was harvested from both implants at 2 positions, anterior contact zone with TiLOOP^®^ and posterior (TiLOOP^®^-free) contact zone with *M. pectoralis*. Samples were placed immediately after withdrawal into sterile boxes stored at 4 °C before transport to the research laboratory. Under sterile conditions, the specimens were stored in (a) DNA/RNA Shield reagent for microbiome sequencing and (b) RPMI medium for primary microbial culture attempts.

Expander exchange with definite implants was performed during reoperation between 6 and 8 months after initial expander implantation. Removed tissue expanders were placed immediately after withdrawal into sterile boxes. For microbiome profiling by NGS, surface swabs were stabilized with DNA/RNA Shield reagent stabilization solution (Zymo Research). For primary culture attempts, we stored surface swabs in RPMI, as well as SMI patches excised with a sterile medical biopsy punch (PFM, ø 8 mm). The device was frozen as well as stored at −80 °C before transport to the research laboratory of the Protein Core Facility for label-free quantitative proteomic analysis. 

### 4.4. The Mass Spectrometry Proteomics Data Source

For detailed descriptions of biological sample preparation, TMT-based quantitative proteomic approach, and label-free quantitative proteomic analysis, see Schoberleitner et al., 2023 [13]. The mass spectrometry proteomics data from plasma, wound bed fluid specimens, and adhesive SMI proteome specimens [13], deposited in the ProteomeXchange Consortium via the PRIDE partner repository with the dataset identifier PXD039840, were subjected to the following.

#### 4.4.1. Identification, Characterization, and Quantification of Differential Common and Topography-Exclusive Wound Bed Proteome

Obtained data from plasma and wound bed fluid specimens were log2 transformed and analyzed for sets of proteins either common and exclusive to SMI 4 µm or SMI 60 µm and associated with both devices in the acute wound as well as interaction with the plasma proteome by Interactivenn [95].

A proportion of 437 proteins of the 895 common plasma-derived wound proteins were identified according to their biological role (investigated in UniProt database) as proteins involved in inflammatory excessive ECM turnover—the inflammatory matrisome. Depending on their annotated role (Uniprot), we functionally annotated these into steps of tissue repair after SMI implantation (step 1: clotting; step 2: inflammation; step 3: repair and fibrogenesis; and step 4: ECM turnover) [21]. Among the 437 inflammatory matrisome proteins, we identified 65 plasma-derived acute-wound-associated inflammatory components in the acute wound that are involved in the antimicrobial humoral response. 

Identified AMPs from both devices were tested for enriched gene ontology (molecular function) [96] as well as by the Search Tool for the Retrieval of Interacting Genes/Proteins (STRING v 11.5) database of physical and functional interactions to analyze the protein–protein interaction (PPI) of selected proteins [97].

Statistical data analysis of the common and topography-exclusive wound proteome was carried out with GraphPad Prism (version 10.1.1). Mean values and standard deviations were calculated for each experimental condition or type of sample. *p*-values between samples were calculated by unpaired *t*-test per protein, with individual variances computed for each comparison, combined with the two-stage linear step-up procedure of Benjamini, Krieger, and Yekutieli. Significance was tested with a two-stage set-up method with a false discovery rate set to 0.01. Proteins were regarded as being differentially expressed when meeting the criteria l2fc ≥ ±1.5 and adjusted *p*-value ≤ 0.01. Heatmaps were generated using the ClustVis [98] tool. Generation of tables was performed with Microsoft Excel 2018 (Microsoft Corporation, Washington, DC, USA).

#### 4.4.2. Identification and Characterization of Common and Topography-Exclusive Adsorbed Wound Bed Proteome on SMI Surface

Obtained abundances from adhesive SMI proteome specimens were analyzed for a common and SMI topography-exclusive adsorbed set of proteins adsorbed by both devices by Interactivenn [95]. Generation of correlation plots was performed using GraphPad Prism (version 10.1.1). The statistical details of experiments are presented in the relevant figure legends. A *p*-value of <0.05 was considered significant. Significance: * *p* < 0.05, ** *p* < 0.01, *** *p* < 0.001, **** *p* < 0.0001, ns = not significant.

### 4.5. Microbiome Profiling by Next-Generation DNA Sequencing

#### 4.5.1. Sample Preparation

Samples collected for microbiome profiling were analyzed using the PrecisionBIOME next-generation DNA sequencing test (Pangea Laboratory, Tustin, CA, USA). Capsular tissue samples preserved in DNA/RNA Shield were processed using the HostZERO Microbial DNA Kit (Zymo Research) to remove human DNA and improve the detection of microbial DNA. Briefly, 20–60 mg of capsular tissue was minced using sterile scalpels before transfer to a ZR BashingBead™ Lysis Tube (2.0 mm). Host depletion was performed following the manufacturer’s protocol for solid tissues using the FastPrep-24™ (MP Biomedicals) bead beater. Microbial DNA was extracted from the host-depleted capsular tissue samples and 1 mL of all other samples preserved in DNA/RNA Shield (e.g., WBF, expander swab, and controls) using the ZymoBIOMICS 96 MagBead DNA Kit (Zymo Research). Sequencing libraries were prepared using the PrecisionBIOME NGS Library Prep Kit (Zymo Research), which includes amplification and barcoding of bacterial 16S rRNA V1-V3 and fungal ITS2 regions. Next-generation DNA sequencing was performed using an Illumina platform, and sequences were analyzed by a proprietary PrecisionBIOME bioinformatics pipeline capable of species-level resolution of bacteria and fungi using the PrecisionBIOME reference sequence database. The input to this pipeline is the raw amplicon (16S or ITS2 region) sequencing reads, and the output is a read count taxonomy table at the species level. To monitor the integrity of the assay, from sample extraction through data analysis, the ZymoBIOMICS Microbial Community Standard (Zymo Research) was used as a positive control. Negative controls included negative extraction controls to monitor reagent contamination as well as those described in Section 4.5.2.

#### 4.5.2. Control Measures for Microbiome Analysis: Normalization of Wound Bed Fluid, SMI Surface, and Capsular Tissue Data to Multiple Negative Controls

In order to prevent the introduction of microbial particles from the environment and ensure the accurate detection and comparison of microbiomes both within individuals and between individuals, we collected several control swabs concurrently with biological sample collection. WBF, SMI surface, and capsular tissue sample data were normalized to negative controls (a) not associated with implants (surgery room atmosphere, DNA/RNA Shield Reagent used for sample stabilization), (b) negative controls exclusive for SMI 60 µm and SMI 4 µm (WBF drainage container valve, “sterile” implant packing, tissue prep area under cell culture flow, scalpel for tissue prep), and negative controls for capsular tissue processing procedure (Host Zero).

#### 4.5.3. Statistical Analyses

Bacterial count data were agglomerated at the species level before normalization against the negative controls, first using the R package decontam version 1.20.0 [99] using the “prevalence” method with a threshold of 0.5. Any taxa remaining which were prevalent in >25% of the combined negative controls were manually removed (an additional 12 taxa). Microbiome data were explored using the R package phyloseq version 1.44.0 [100] and visualized using the ggplot2 library version 3.4.4 [101]. Alpha diversity was calculated using the observed richness method at the family taxonomic level using the phyloseq ‘estimate richness’ function, and statistical differences were explored between the SMI 4 µm and SMI 60 µm expanders and specimen types by pairwise Wilcoxon signed-rank test using the R stats package version 4.3.1. Beta diversity was investigated at the family level using the phyloseq ‘ordinate’ function, with a principal coordinate analysis (PCoA) constructed from a Bray–Curtis dissimilarity matrix. Permutational multivariate analysis of variance (PERMANOVA) was calculated using the ‘adonis2’ function from the R package vegan (version 2.6.4) to assess the impact of the covariate, expander type, on the variability of bacterial community composition.

Obtained data (cell counts) from wound bed fluid, capsular tissue specimens, and expander swabs were visualized by principal component analysis with the ClustVis tool [98]. In the latter, unit variance scaling was applied to rows and SVD with imputation was used to calculate principal components. 

The total of all counts in all WBF, expander surface swabs, and capsular tissue samples accumulated around SMI 60 µm was compared to a total of counts detected around SMI 4 µm. Heatmaps were generated using the ClustVis tool. Original values (read counts) were ln(x + 1)-transformed. Rows were centered and unit variance scaling was applied to rows. Both rows and columns were clustered using correlation distance and average linkage.

### 4.6. Validation of Sequencing Data: Cultivation and MALDI TOF Identification of Skin Swab Samples from Surgical Incision Sites and Proximal Regions

To validate the sequencing data, we confirmed our results through cultivation and MALDI TOF identification of skin swab samples collected from the incision site (under bust), proximity of the incision site (armpit), and, as control, the neck. Sample collection was performed during regular postoperative controls at our department with swabs stored in Amies gel during transport (Sarstedt). Swabs of clinical bacterial isolates were cultivated on SAB agar for 24 h at 37 °C and quantified subsequently by MALDI TOF mass spectrometry.

### 4.7. Microbial Culture and Biofilm Formation on SMI Surfaces In Vitro

#### 4.7.1. SMI Material and Sterilization

In the in vitro part of our study, we investigated three silicone implants—Smooth MENTOR^®^ (REF: 350-2254BC; S/N: 9462602-031), SmoothSilk^®^ Motiva Ergonomix^®^ (REF: M15.00-07.80RE; S/N: SE21010477), and MENTOR^®^ CPX™4 (REF: 354-9123; LOT: 7668141; SN: 7668141-006), provided and sterilized by Establishment Labs (Costa Rica). Smooth MENTOR^®^ exhibited an average surface roughness (Ra) of 1 µm [102], referred to as ‘smooth’ henceforth. SmoothSilk^®^ Motiva Ergonomix^®^, with an average surface roughness of 4 µm [102,103,104], was designated as SMI 4 µm, while MENTOR^®^ CPX™4, possessing a roughness of approximately 60 µm [103,104], was labeled as SMI 60 µm. Patches were extracted from the silicone implants using an 8.0 mm punch (KAI MEDICAL, REF: BP-80F; LOT: 10H22) and subsequently placed in 24-well plates (Greiner bio-one CELLSTAR^®^, Cat. No. 662-102; LOT: E21043KF). 

#### 4.7.2. Biofilm Formation and Bacteria

In this investigation, we studied the biofilm-forming strains *Staphylococcus aureus ATCC 6538* and *Staphylococcus epidermidis ATCC 12228*. For biofilm development, a bacterial suspension with a concentration of 5 × 10^7^ CFU/mL in RPMI was prepared using the McFarland standard for each *Staphylococcus* strain (Appendix A). Subsequently, 700 µL of this solution was added to each well, ensuring complete coverage of the silicone patches. Negative controls included patches without bacteria in BenchStableTM RPMI + GlutaMAXTM-I (1X) (Ref. No. A41923-01, Lot. No. 2307202) media and wells with RPMI only. All samples were incubated for 48 h at 37 °C on a shaker set at 300 RPM as previously described in [82,105]. 

#### 4.7.3. Biofilm Dissolution and Quantification

To dissolve the biofilm, individual silicone disks underwent three washes with PBS to remove planktonic and loosely attached cells. The disks were then transferred to separate 2 mL tubes (Safe-Lock Tubes 2.0 mL 500 Eppendorf Tubes^®^, LOT: H179833O). Following this, the patches were covered with 700 µL of fresh RPMI and subjected to five minutes of ultrasound treatment using an ultrasound bath (Elma^®^ Transsonic 570, No. V906738058), aimed at dissolving any formed biofilm. Following the ultrasound bath, 50 µL from each tube was plated in triplicates on MHA plates. The freshly plated MHA plates were incubated for 48 h at 37 °C. After incubation, we counted the CFU. The CFUs for each disk were extrapolated to a concentration in CFU in ml for statistical analysis. A mixed-effects analysis of variance (ANOVA) was conducted on the differences in the colonization of the different silicone surfaces using GraphPad Prism 8.0.1 (GraphPad Software, Inc., La Jolla, CA, USA, 2007). Pairwise comparisons were carried out with the use of Tukey’s multiple comparisons test; for the comparison of more than two groups a one-way ANOVA was calculated. The ANOVA included experiment as the random effect, and species and implant type were treated as fixed effects. Due to indications of an interaction between species and implant type from interaction plots and tests, the differences for each species were analyzed independently to explore the interaction further. All experiments were performed in biological triplicates. Correlations were calculated by Pearson analysis.

#### 4.7.4. Scanning Electron Microscopy 

Implant morphology and topography were characterized by scanning electron microscopy (SEM). To visualize biofilm formation on SMI patches, after incubation with *S. epidermidis* and *S. aureus*, individual silicone patches were washed three times in PBS. Next, the patches went through a sequence of increasing alcohol concentrations for dehydration, beginning with 50%, then 70%, 80%, and eventually 99% ethanol. After air-drying for a minimum of 5 min, the patches were mounted on aluminum pins and sputtered with gold using an AGAR sputter coater (P5240-012) for 45 s at 30 mA. Representative images of each surface were taken using a JSM-6010LV scanning electron microscope, Jeol GmbH, Freising, Germany.

### 4.8. Bacterial Biofilm Gene Expression and PBMC Inflammatory Gene Expression Using Real-Time qPCR 

#### 4.8.1. Cell Isolation and In Vitro Culture with Biofilm-Associated SMI Surfaces

PBMC were isolated from whole blood using the Ficoll-Paque™ (VWR) standard density gradient centrifugation method. An amount of 10^6^ PBMC was seeded on each surface and cultured in 24-well plates (Greiner bio-one CELLSTAR^®^, Cat.-No.: 662-102; LOT: E21043KF) for 24 h in 1000 µL BenchStableTM RPMI + GlutaMAXTM-I (1X) (Ref. No. A41923-01, Lot. No. 2307202) without antibiotics supplementation at 37 °C in 5% CO_2_. As control, cells were seeded onto SMI patches without bacterial inoculation or/and directly onto a tissue culture polystyrene well plate Greiner bio-one CELLSTAR^®^, Cat.-No.: 662-102; LOT: E21043KF). After incubation time, supernatant was subjected to RNA isolation for pro-inflammatory gene expression analysis.

#### 4.8.2. Reverse Transcription Real-Time Quantitative PCR (RT-qPCR)

To test the immunoreactivity potential of biofilm-covered SMI patches in vitro, we used *S. aureus ATCC 25923* and *S. epidermidis ATCC 12228* as biofilm-forming strains. Following the ultrasound bath, 650 µL from each tube was used for RT-qPCR analyses to evaluate expression levels of genes responsible for biofilm formation on *S. epidermidis* and *S. aureus* biofilm-covered smooth, SMI 4 µm, and SMI 60 µm patches.

To analyze PBMC gene expression of pro-inflammatory and profibrotic markers as a response to the PBMC cocultivation and exposure to SMI-covered Staphylococcae biofilms, we used 1000 µL supernatant of cocultivated PBMC and biofilm-covered SMI surfaces. 

Briefly, all biological samples were frozen in triplicates at −80 °C immediately after collection and processed for further analysis. Total RNA was extracted using TRI Reagent^®^ (Sigma Aldrich, Merck KGaA, Darmstadt, Germany) followed by RNA purification with Monarch RNA Clean up Kit (NEB) and cDNA synthesis with LunaScript RT SuperMix Kit (NEB). qPCR was performed in triplicate using Luna^®^ Universal qPCR Master Mix (NEB) with 25 ng cDNA and 0.4 μM of target-specific primers in a Bio-rad CFX instrument (Bio-Rad, Hercules, CA 94547, USA). Primer sequences are available upon request. For biofilm gene expression, transcripts were normalized to bacterial *16S rRNA*. For PBMC inflammatory gene expression, transcripts were normalized to *MT-ATP6* and *B2M* as previously described in [37]. Transcript levels were expressed relative to patches without bacterial inoculation, along with NaCl-based cultivation as control. The 2−ΔΔCt values were calculated and statistical analysis was performed by unpaired Student’s *t*-test (GraphPad Prism 8.2.1).

### 4.9. Statistics

Graphing and statistics were performed using GraphPad Prism software v.8.2.1. The statistical details of the experiments are presented in the relevant figure legends. The level for statistical significance was set at *p* ≤ 0.05 for all statistical tests and significant differences were marked (* *p* ≤ 0.05, ** *p* ≤ 0.01, *** *p* ≤ 0.001, **** *p* ≤ 0.0001, ns—not significant).

### 4.10. Ethics Statement

All biological samples from donors, including blood, wound bed fluid, and removed tissue expanders, were acquired following informed written consent from the participants (Ethics Committee of the Medical University of Innsbruck, approval number 1325/2019). The collection and usage of these samples adhered to the guidelines outlined in §40, Section 3 of the Medical Devices Act, with approval granted by the Austrian Federal Office for Safety in Health Care (approval number 13340962). 

## 5. Conclusions

Our study provides a basis for understanding the multifaceted interactions between SMIs, microbial dynamics, and the antimicrobial immune response, unraveling crucial aspects that influence clinical outcomes in breast implant-based reconstructions. The foreign body response to an SMI initiates an inflammatory reaction, forming a complex interplay between immune cells, antimicrobial substances, and the implant surface. Proteomic investigations revealed a diverse array of proteins associated with the immune response, inflammatory processes, and wound healing in proximity to SMIs. Reduction of implant surface roughness to Ra 4 µm emerged as a promising approach to mitigate detrimental immune reactions and curb excessive fibrosis, as evidenced by the identification of specific antimicrobial proteins. Microbiome analysis, by 16S rRNA quantification as well as cultivation, further emphasized the impact of surface topography on microbial composition and abundance, with higher microbial diversity observed around rougher implants. Additionally, in vitro experiments illustrated the significant influence of SMI surface texture on bacterial adhesion, growth, colonization, and biofilm formation. The heightened expression of biofilm markers in response to textured surfaces substantiated the in vivo findings. Notably, our investigation into skin microbiota transfer during surgery highlighted potential associations between specific microbial species and the surgical site. While providing valuable insights, this study underscores the need for larger-scale research to comprehensively understand microbial dynamics and improve procedures to avoid microbial contamination and their implications for surgical outcomes. Ultimately, our findings contribute to the ongoing efforts to develop implants with optimized properties to minimize complications in clinical settings due to infections.

## Figures and Tables

**Figure 1 ijms-25-03163-f001:**
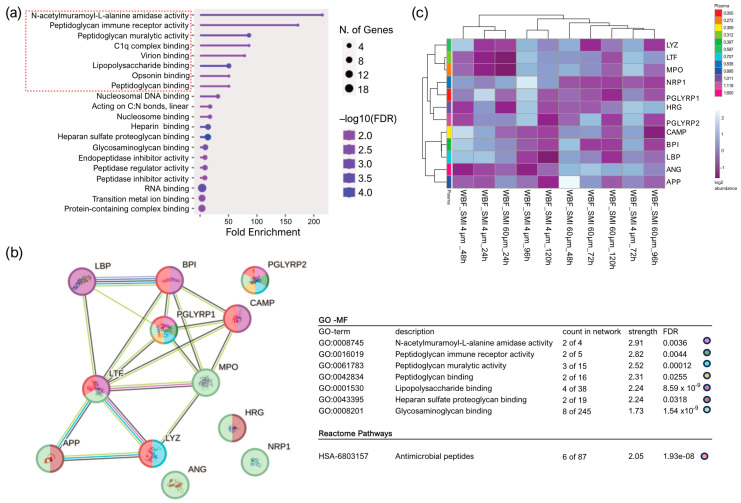
Acute plasma-derived antimicrobial response in wound fluid surrounding SMI 24–120 h post-op. Plasma-derived wound proteome. (**a**) GO molecular function enrichment; (**b**) protein–protein interaction regulatory network by K-means clustering based on STRING database (Reference NCBI taxonomy Id: 9606); and (**c**) heatmap analysis of the 12 interacting AMPs in the plasma-derived wound proteome. Original values (log2 protein abundance) are ln (x + 1)-transformed. Rows are centered; unit variance scaling is applied to rows. Both rows and columns are clustered using correlation distance, average method, and tightest cluster first tree ordering: 12 rows, 10 columns.

**Figure 2 ijms-25-03163-f002:**
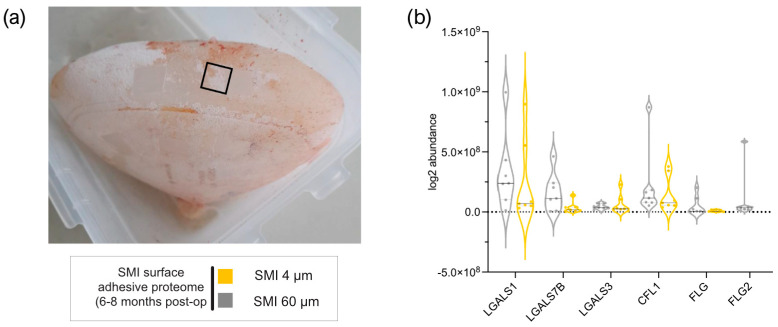
Plasma-derived antimicrobial response proteins adhered to SMI surface 6–8 months post-op. (**a**) Photo of removed SMI 60 µm. Black square marks testing area. (**b**) Protein abundance of different AMPs adhered to SMI 4 µm and/or SMI 60 µm surface.

**Figure 3 ijms-25-03163-f003:**
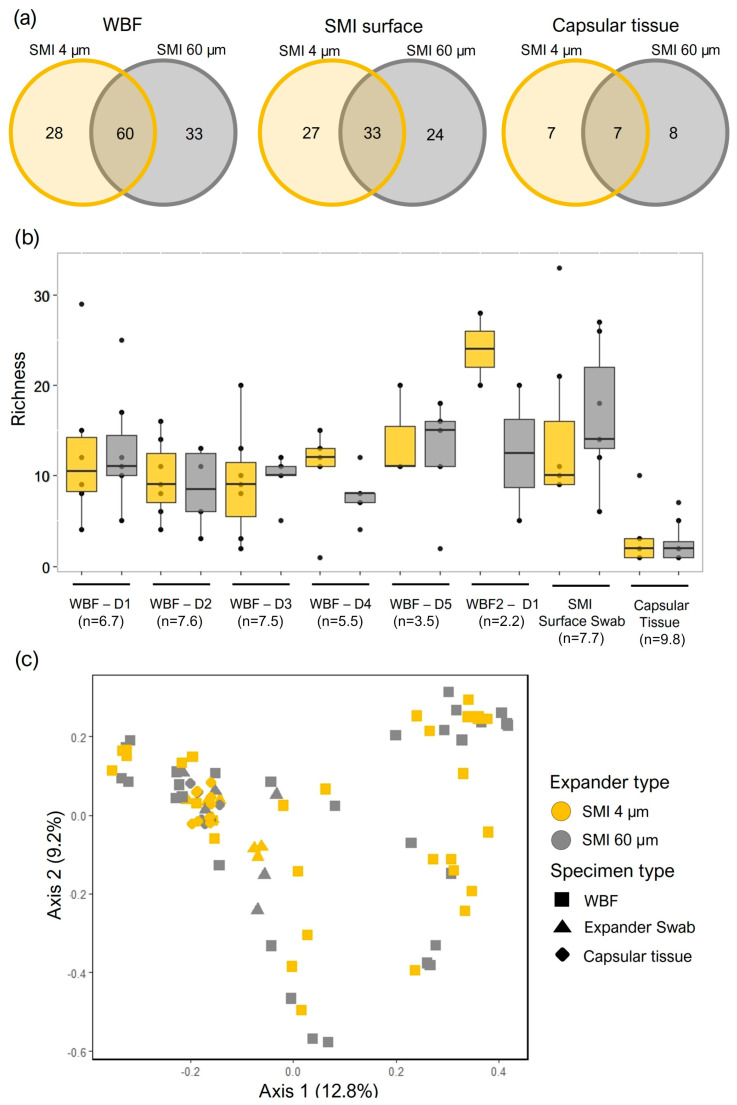
Analysis of the bacterial communities by NGS. (**a**) Venn diagram showing the counts of bacteria at the family level identified in the collected wound bed fluid (WBF), SMI surface, and capsular tissue, originating from the SMI 4 µM and SMI 60 µM tissue expanders. (**b**) Observed alpha diversity measured as taxa richness at the family level of the wound drainage fluid in the first 5 days following the initial surgery (tissue-expander implantation, WBF D1-5) and in the day following the second surgery (definitive implant implantation, WBF2 D1), the SMI surface swab, and the capsular tissue. No statistically significant differences computed by pairwise Wilcoxon signed-rank test were observed between the SMI 4 µM and SMI 60 µM tissue-expander specimens. (**c**) Principal coordinate analysis of beta diversity based on Bray–Curtis dissimilarity. Plot shows first two principal coordinates, with each point representing a specimen, colored according to expander type and shaped according to specimen type. Permutational analysis of variance (PERMANOVA) revealed no significant differences in bacterial community composition between the two expander types for each specimen type.

**Figure 4 ijms-25-03163-f004:**
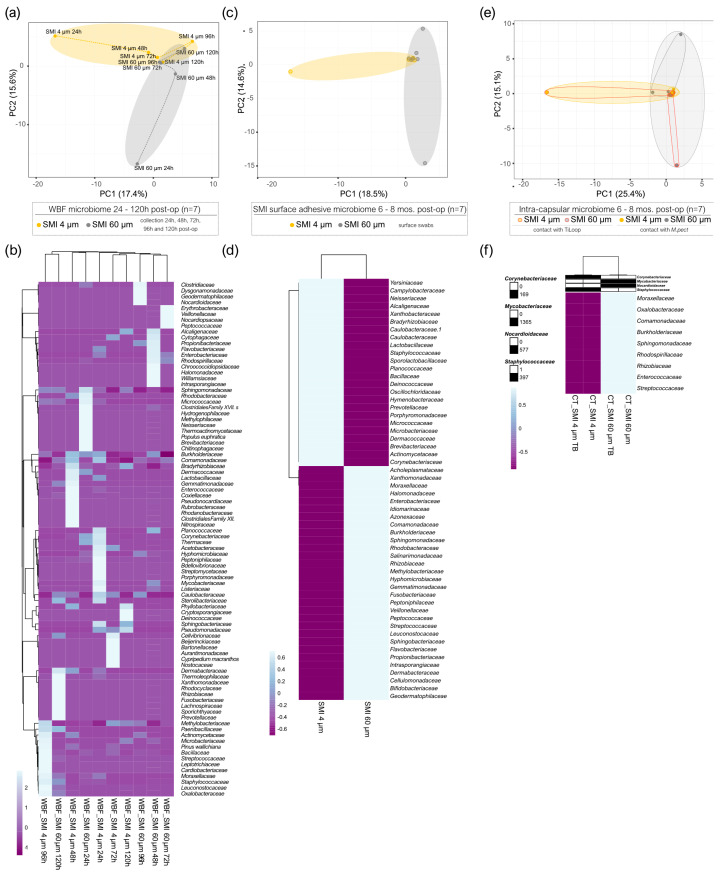
Bacterial colonization (**a**,**b**) the first five days post-op in the acute wound (WBF 24–120 h post-op) and (**c**,**d**) adhered to SMI surface as well as incorporated into the capsule (**e**,**f**) 6–8 months (mos.) post-op. (**a**,**c**,**e**) PCA was obtained with cell counts of the microbial sequences found in all samples of WBF, SMI surface swabs, and capsular tissue. Unit variance scaling is applied to rows; SVD with imputation is used to calculate principal components. X and Y axes show principal component 1 and principal component 2 that explain (**a**) 17.4% and 15.6% (N = 10 data points) of total variance within the WBF microbiome formed around two different textured SMIs 24–120 h post-op, (**c**) 14.6% and 18.5% (N = 14 data points) of total variance within the adhered microbiome on two different textured SMI surfaces 6–8 M post-op, and (**e**) 15.1% and 25.4% (N = 22 data points) of the total variance of the intracapsular microbiome encapsulating two different SMI surfaces, respectively. (**b**,**d**,**f**) Heatmap analysis of microbial composition in WBF 24–120 h post-op, on SMI surface, and integrated into capsule 6–8 M post-op. Original values (counts) are ln (x + 1)-transformed. Rows are centered; unit variance scaling is applied to rows. Both rows and columns are clustered using correlation distance and average linkage; (**b**) 244 rows, 10 columns; (**d**) 54 rows, 2 columns; (**f**) 14 rows, 4 columns.

**Figure 5 ijms-25-03163-f005:**
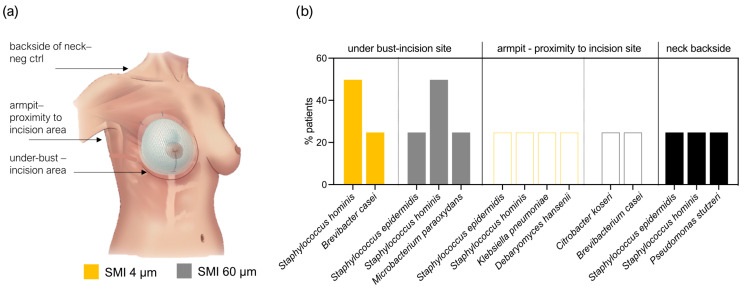
Culture-based detection of skin microbiota at the operative incision sites and the back of the neck (negative control). (**a**) Schematic representation of swab sample collection sites. (**b**) Swabs were cultivated on SAB agar and subjected to MALDI TOF detection.

**Figure 6 ijms-25-03163-f006:**
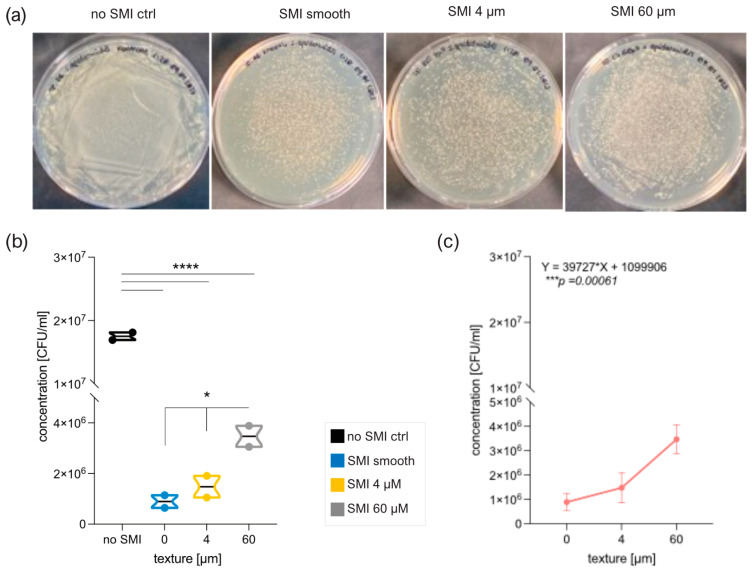
The growth and colonization of *S. epidermidis* are significantly influenced by silicone (the silicone patch) and SMI surface topography. (**a**) Incubated *S. epidermidis* on MHD-Agar after 24 h at 37 °C ON positive control without silicone patch, smooth SMI, SMI 4 µm, and SMI 60 µm. (**b**) Truncated violin plot of *S. epidermidis* growth quantification (colony forming units per ml (CFU/mL)). One-way ANOVA: (silicone patch) F (3, 4) = 318.6, **** *p* < 0.0001; (patch topography) F(2, 3) = 13.15, * *p* = 0.0327. (**c**) Correlation analysis of *S. epidermidis* concentration and SMI surface roughness. Pearson r = 0.9873, ^ns^
*p* = 0.1015. Simple linear regression equation is denoted above the corresponding panel (slope significantly nonzero: *** *p* = 0.00061).

**Figure 7 ijms-25-03163-f007:**
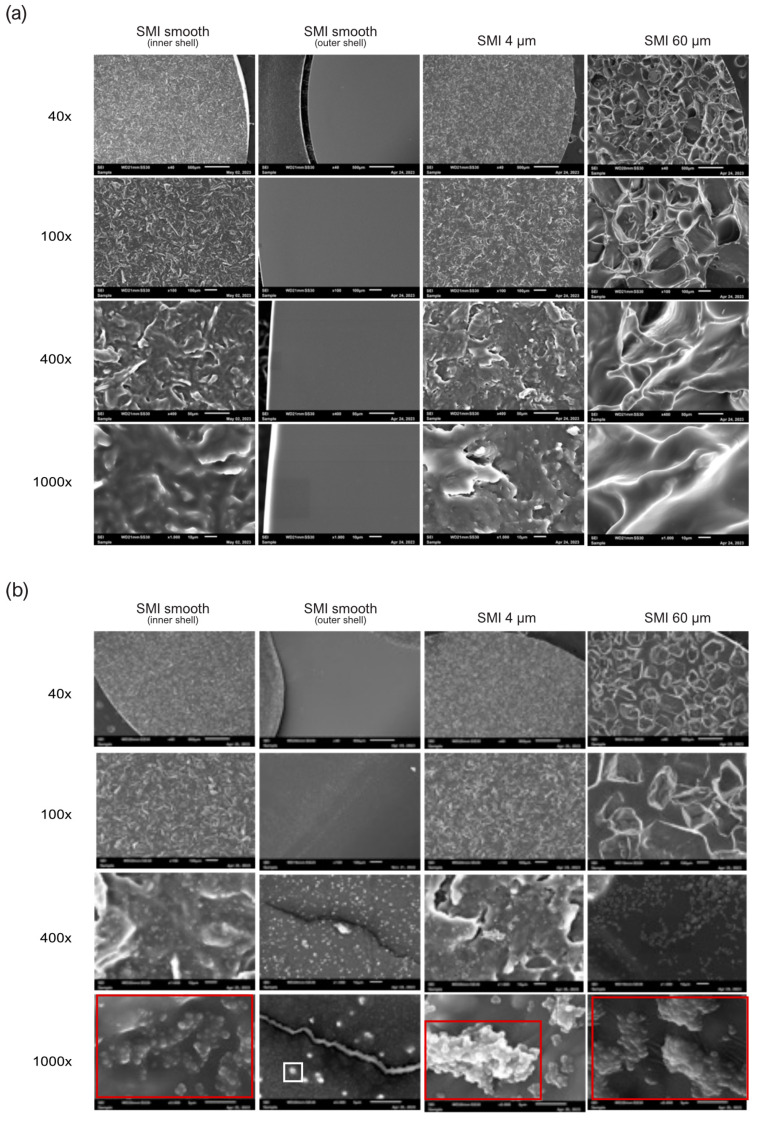
Biofilm formation of *S. epidermidis* is influenced by silicone (the silicone patch) and SMI surface topography. (**a**) Microscopic images of negative control silicone surfaces without microbial colonization/biofilm. No growth detected. (**b**) Microscopic images of the silicone patches inoculated with *S. epidermidis* on the inner side of the smooth implant, the outer side of the smooth implant, SMI 4 µm, and SMI 60 µm. The rows show different magnifications, from top to bottom: 40×, 100×, 4000×, 1000×. White square marks a single bacterium, red square surrounds the biofilm.

**Figure 8 ijms-25-03163-f008:**
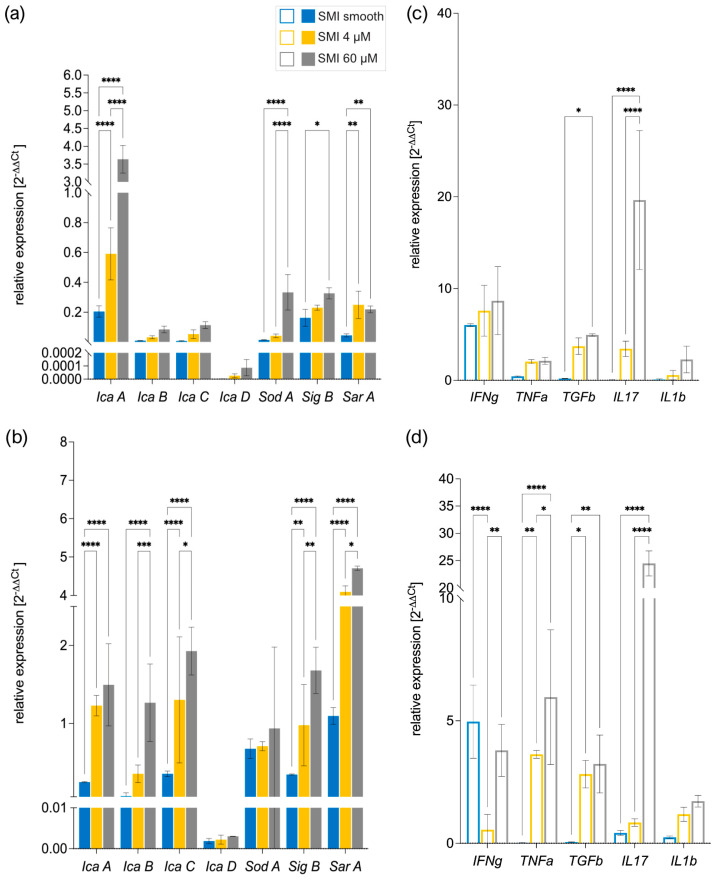
RT-qPCR analysis of (**a**,**b**) bacterial biofilm marker *IcaABCD*, *SodA*, *SigB*, *SarA* gene expression of (**c**) *S. epidermidis* and (**b**) *S. aureus biofilm-covered* SMI patches cultivated for 48 h at 37 °C; and (**c**,**d**) PBMC-expressed pro-inflammatory immune response markers *IFNγ*, *TNFα*, *TGFβ*, *IL17*, and *IL1b* on (**c**) *S. epidermidis* and (**d**) *S. aureus biofilm-covered* SMI patches cultivated with PBMCs for 24 h at 37 °C. Transcript levels were normalized to (**a**,**b**) bacterial *16S rRNA* and (**c**,**d**) human *16S rRNA* (as well as *MT-ATP6*, *B2M*) expressed relative to SMI patches not inoculated with bacteria or cultured with NaCl as control. Statistical significance was determined by 2-way ANOVA: SMI surface topography main effect: (**a**) F(2, 30)  =  26.74, **** *p* < 0.0001; (**b**) F(2, 30) =  166.9, **** *p* < 0.0001; (**c**) F(2, 105) = 44.1, **** *p* < 0.0001; and (**d**) F(2, 105) = 139.7, **** *p* < 0.0001; gene expression main effect: (**a**) F(4, 30) = 16.98, **** *p* < 0.0001; (**b**) F(4, 30) = 62.86, **** *p* < 0.0001; (**c**) F(6, 105) = 490.6, **** *p* < 0.0001; and (**d**) F(6, 105) = 145.0, **** *p* < 0.0001; SMI surface topography × biofilm/cytokine gene expression interaction effect: (**a**) F(8, 30) = 9.575, **** *p* < 0.0001; (**b**) F(8, 30) = 85.42, **** *p* < 0.0001; (**c**) F(12, 105) = 282.1, **** *p* < 0.0001; and (**d**) F(12, 105) = 18.94, **** *p* < 0.0001 and Tukey post hoc test (significance denoted in graph). The level for statistical significance was set at ^ns^
*p* > 0.05, * *p* < 0.05, ** *p* < 0.002, *** *p* < 0.0002, and **** *p* < 0.0001 for all statistical tests (inter- and intraindividual comparison; *n* = 4 × topography × 3x biological replicates = 12).

**Table 1 ijms-25-03163-t001:** Inclusion and exclusion criteria for the Expander-Immunology Trial.

Inclusion Criteria	Exclusion Criteria
Female sex	Severe coagulation disorder, representing a potential contraindication for the elective surgery
Age > 18 years	Rheumatic disease accompanied by obligatory intake of immunomodulating therapeutic agents
High-risk family history for breast and/or ovarian cancer and/or *BRCA1/2* gene mutation carrier	Severe renal functional disorder: renal insufficiency status iv or v (estimated glomerulary filtration rate (gfr) < 30 mL/min)
Planned bilateral mastectomy with simultaneous breast reconstruction	Active hematological or oncological disease
	HIV infection
	Hepatitis infection
	Pregnancy or breast-feeding
	Intake of anti-inflammatory drugs
	Carrier of silicone implants (e.g., gastric banding, mammary implants)
	Severe coagulation disorder, representing a potential contraindication for the elective surgery
	Rheumatic disease accompanied by obligatory intake of immunomodulating therapeutic agents

## Data Availability

The mass spectrometry proteomics data have been deposited to the ProteomeXchange Consortium (http://proteomecentral.proteomexchange.org) via the PRIDE partner repository with the dataset identifier PXD039840 and are publicly available. Bacterial read count data resulting from NGS analysis are available in Appendix A. The data presented in this study are available on request from the corresponding author due to data privacy protection. The trial basic summary results have been deposited within ClinicalTrials.gov (ID: NCT05648929) registry and are publicly available as of the date of publication.

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
