# Peer review of "Surface Topography, Microbial Adhesion, and Immune Responses in Silicone Mammary Implant-Associated Capsular Fibrosis"

_ijms, 2024, doi:10.3390/ijms25063163_

Round 1

Reviewer 1 Report

Comments and Suggestions for Authors

The study conducted by Schoberleitner et al. presents a comprehensive investigation into the intricate interplay between silicone mammary implant (SMI) surface topography, wound proteome dynamics, microbial colonization, and host immune responses in breast reconstruction. The findings shed light on critical aspects of implant-associated complications and provide insights for optimizing clinical practices in breast reconstruction surgery.

Minor Revision Suggestions:

1.      Clarification on Methodology: Provide additional details regarding the methodology employed for next-generation DNA sequencing (NGS) analysis, including sample preparation, sequencing protocols, and data processing pipelines. This will enhance the reproducibility and clarity of the microbiome analysis section.

2.      Discussion of Limitations: Acknowledge potential limitations of the study, such as sample size variations, inherent patient-specific factors influencing microbial composition, and the need for further longitudinal studies to validate findings over an extended follow-up period.

3.      Incorporate Future Directions: Consider including a section discussing potential avenues for future research, such as exploring the impact of antibiotic prophylaxis, surgical techniques, or implant coatings on microbial colonization and host responses to further optimize clinical outcomes in breast reconstruction.

Overall, the study represents a significant advancement in understanding the complex interactions between SMI topography, microbial colonization, and host immune responses in breast reconstruction. Addressing the suggested revisions will further strengthen the manuscript and enhance its impact on the field.

Author Response

Dear Reviewer,

Thank you for your thoughtful comments and recommendations. For a detailed point-by-point response and performed revision of the manuscript, please see the attachment.

Reviewer 2 Report

Comments and Suggestions for Authors

The authors conducted a valuable study on the impact of SMI surface topography on host antimicrobial responses, wound proteome dynamics, and microbial colonization.

The abstract is good, but the conclusion section can be improved.

The number of keywords is huge. Does the journal have no limitation in this regard?

The introduction provides sufficient background and includes all relevant references. However, some more recent and relevant references are needed. Also, the introduction section seems to be too long. It would be better to be a little more concise.

Please re-order the method section. For example: transfer the "Ethics Statement" to the end of the method section.

Please write the sample size determination method.

Please add the inclusion and exclusion criteria.

The quality of the figures is low.

Discussion can improve and expand using more recent and relevant references. 

Author Response

Dear Reviewer,

We sincerely appreciate your thorough review and constructive feedback on our manuscript. We have diligently addressed each of your concerns and made the necessary revisions to enhance the clarity, validity, and overall quality of our work. Please see the attachment for a detailed summary of the modifications made in response to your valuable comments.

Reviewer 3 Report

Comments and Suggestions for Authors

The current work aims to investigate the surface topography, microbial Adhesion, and immune responses in silicone mammary implant-associated capsular fibrosis. Although the topic is interesting in its scientific field, there are some issues that require the authors’ attention to improve the quality of this particular manuscript before further consideration for publication in a high-quality journal “IJMS”.

Specific comments:

1.         There are no statistical labels on the data presentation in Figures 1 and 2. Please improve.

2.         The authors stated an immediate inflammatory storm with wound bed fluid (WBF) proteomes formed around SMI 4 μm at 24h and 48h post-operation and around SMI 60 μm at 24h post-operation. However, the lack of relevant evidences is disadvantageous to support this claim. Please provide these important experimental data.

3.         It is necessary to explore the relationship between specific surface topographies and their impact on immune responses and microbial adhesion. Please improve.

4.         This study aims to compare the surface morphology of two types of SMI. However, the authors did not provide surface morphological data of SMI by AFM. Please improve.

5.         Figure 6a shows Staphylococcus epidermidis adhesion and colonization on all tested silicone surfaces. However, it is difficult to see any obvious differences from the imaging data. The authors are highly recommended to present the experimental outcomes at different time points to examine bacterial adhesion and colonization behaviors.

6.         The description of electron microscopic imaging data is too brief to convey clear information to the readers. The authors should improve the discussion of Figure 6d and explain the differences between the studied groups.

7.         As stated by the authors, this raises the possibility of microbial colonization and biofilm formation on the implant surface. However, this important claim was not supported by any documented reference. If possible, the authors may consider the inclusion of this supportive case study (DOI: 10.1016/j.cej.2019.123913) in the reference list to enrich the article content and balance scientific viewpoint.

Author Response

Dear Reviewer,

We sincerely appreciate your comprehensive review and valuable feedback on our manuscript. We have conscientiously attended to each of your concerns and implemented necessary revisions to improve the clarity, validity, and overall quality of our work. Please see the attachment for a comprehensive summary of the modifications made in response to your insightful comments.

Round 2

Reviewer 3 Report

Comments and Suggestions for Authors

The authors’ responses to comments #2, #3, #4, and #5 are unreasonable and should be further addressed. Please carefully improve these unsolved issues by providing necessary experimental evidences.

Comment #2: The authors stated an immediate inflammatory storm with wound bed fluid (WBF) proteomes formed around SMI 4 μm at 24h and 48h post-operation and around SMI 60 μm at 24h post-operation. However, the lack of relevant evidences is disadvantageous to support this claim. Please provide these important experimental data.

Comment #3: It is necessary to explore the relationship between specific surface topographies and their impact on immune responses and microbial adhesion. Please improve.

Comment #4: This study aims to compare the surface morphology of two types of SMI. However, the authors did not provide surface morphological data of SMI by AFM. Please improve.

Comment #5: Figure 6a shows Staphylococcus epidermidis adhesion and colonization on all tested silicone surfaces. However, it is difficult to see any obvious differences from the imaging data. The authors are highly recommended to present the experimental outcomes at different time points to examine bacterial adhesion and colonization behaviors.

Author Response

Dear Reviewer, we appreciate the thorough evaluation of our manuscript. Please see the attachment for a point-by-point response to your concerns.

Round 3

Reviewer 3 Report

Comments and Suggestions for Authors

The revised version has adequately addressed most of the critiques raised by this reviewer.